# A comparative field evaluation of six medicine quality screening devices in Laos

Céline Caillet[1,2,3,4‡]*, Serena Vickers[1,2,3‡], Stephen Zambrzycki[5], Facundo
M. Fernández[5], Vayouly Vidhamaly[1,2,3], Kem Boutsamay[1,2,3],
Phonepasith Boupha[1,2,3], Pimnara Peerawaranun[4], Mavuto Mukaka[2,4], Paul
N. Newton[1,2,3,4]

**1** Lao-Oxford-Mahosot Hospital-Wellcome Trust Research Unit, Microbiology Laboratory, Mahosot Hospital, Vientiane, Lao PDR, **2** Centre for Tropical Medicine and Global Health, Nuffield Department of Medicine, University of Oxford, Oxford, United Kingdom, **3** Infectious Diseases Data Observatory (IDDO)/WorldWide Antimalarial Resistance Network (WWARN), University of Oxford, Oxford, United Kingdom, **4** Mahidol Oxford Tropical Medicine Research Unit (MORU), Faculty of Tropical Medicine, Mahidol University, Bangkok, Thailand, **5** School of Chemistry and Biochemistry, Georgia Institute of Technology, Atlanta, Georgia, United States of America

‡ These authors share first authorship on this work.
* celine.caillet@iddo.org

**Data Availability Statement:** All relevant data are within the manuscript and its Supporting Information files.

## Abstract

### Background

Medicine quality screening devices hold great promise for post-market surveillance (PMS). However, there is little independent evidence on their field utility and usability to inform policy decisions. This pilot study in the Lao PDR tested six devices' utility and usability in detecting substandard and falsified (SF) medicines.

### Methodology/principal findings

Observational time and motion studies of the inspections by 16 Lao medicine inspectors of 1) the stock of an Evaluation Pharmacy (EP), constructed to resemble a Lao pharmacy, and 2) a sample set of medicines (SSM); were conducted without and with six devices: four handheld spectrometers (two near infrared: MicroPHAZIR RX, NIR-S-G1 & two Raman: Progeny, Truscan RM); one portable mid-infrared spectrometer (4500a), and single-use paper analytical devices (PAD). User experiences were documented by interviews and focus group discussions.

Significantly more samples were wrongly categorised as pass/fail with the PAD compared to the other devices in EP inspections ($p < 0.05$). The numbers of samples wrongly classified in EP inspections were significantly lower than in initial visual inspections without devices for 3/6 devices (NIR-S-G1, MicroPHAZIR RX, 4500a). The NIR-S-G1 had the fastest testing time per sample (median 93.5 sec, $p < 0.001$). The time spent on EP visual inspection was significantly shorter when using a device than for inspections without devices, except with the 4500a, risking missing visual clues of samples being SF. The main user errors were the selection of wrong spectrometer reference libraries and wrong user

**Funding:** This work has been co-funded by the Regional Malaria and other Communicable Disease Threats Trust Fund, which has been co-financed by the Government of Australia (Department of Foreign Affairs and Trade); the Government of Canada (Department of Foreign Affairs, Trade and Development); and the Government of the United Kingdom (Department for International Development). The grant (RETA 8763) was managed by the Asian Development Bank (https://www.adb.org/) and awarded to PNN. Additional support was provided by Wellcome Trust Grant N° 202935/Z/16/Z (https://wellcome.org/), awarded to PNN. The funders had no role in study design, data collection and analysis, decision to publish, or preparation of the manuscript. For the purpose of Open Access, the author has applied a CC BY public copyright licence to any Author Accepted Manuscript version arising from this submission.

**Competing interests:** The authors have declared that no competing interests exist.

interpretation of PAD results. Limitations included repeated inspections of the EP by the same inspectors with different devices and the small sample size of SF medicines.

## Conclusions/significance

This pilot study suggests policy makers wishing to implement portable screening devices in PMS should be aware that overconfidence in devices may cause harm by reducing inspectors' investment in visual inspection. It also provides insight into the advantages/limitations of diverse screening devices in the hands of end-users.

### Author summary

Substandard and falsified (SF) medicines threaten the lives of millions of people, especially where pharmaceutical legislation and regulation are limited. Screening for SF medicines in supply chains ('post-market surveillance') by medicine inspectors is crucial but currently relies only on subjective visual inspection of medicines in most countries. Many innovative portable screening technologies now exist and could be key additional assets to the current practice, but none have been extensively evaluated for medicine quality post-market surveillance. We assessed the utility and usability of six screening devices in the hands of Lao medicines inspectors in a pharmacy constructed to resemble a Lao pharmacy. Five spectrometers showed promising accuracies to identify falsified medicines, but difficulties to correctly set them up before running tests were observed. Reading and interpreting colour barcodes of the 'paper analytical cards'–a lab-on-a-chip test by inspectors were difficult, leading to lower accuracy than with spectrometers. The study suggests that overconfidence in devices may cause harm by reducing inspectors' investment in visual inspection—a crucial step to identify falsified medicines. Advantages/limitations of the devices are also documented to inform policy.

## Background

According to a recent World Health Organization (WHO) report, ~10.5% of medical products circulating in low- and middle-income countries (LMICs) are either substandard or falsified (SF) [1]. Falsified medicines are the result of criminal activity, purporting to be genuine, authorized medicines, but are deliberately and fraudulently mislabelled with respect to identity and/or source [2]. Substandard medicines are 'authorized medical products that fail to meet either their quality standards or their specifications, or both' [2].

Currently, national Medicines Regulatory Authorities (MRA) medicine inspectors in LMICs performing post-marketing surveillance (PMS) largely rely only on their own senses and knowledge to detect circulating SF [3]. A plethora of portable analysis screening tools have been developed over the last decade [4,5], allowing some degree of objective analysis of medicines in the 'field'. However, there are enormous key gaps regarding the evidence-base to inform national MRAs of the optimal choice of device to detect SF medical products [4,6].

This is the third paper in the Collection 'A multi-phase evaluation of portable screening devices to assess medicines quality for national Medicines Regulatory Authorities', evaluating devices in Laos. Six devices deemed 'field suitable' in the laboratory evaluation phase were evaluated in the hands of medicine inspectors from the Lao Bureau of Food and Drug Inspection (BFDI) of the Ministry of Health [7]. Inspectors of medicines quality in Laos typically

undertake routine inspection of pharmacies bi-annually, focusing on adherence to legislation and drug registration. Occasionally, medicines are purchased from a selection of pharmacies for screening using the Minilab [8]. All samples which fail Minilab screening, and a further 10% of those which pass, are sent to the National Center for Food and Drug Analysis (NCFDA) [previously 'Food and Drug Quality Control Center'], Vientiane, for pharmacopeial testing.

We aimed to assess the utility and usability of six portable screening devices in the hands of Lao medicine inspectors for inspection in a simulated Evaluation Pharmacy.

## Methods

Six devices and the Minilab, in line with its current use in Laos, were evaluated (Table 1). An outline of the different steps is given in Fig 1.

### Study setting

An evaluation pharmacy (EP) was fashioned at Mahosot Hospital, Vientiane to resemble a Lao Class 2 pharmacy [18,19] and stocked with genuine and falsified field-collected medicines (FCM) stated to contain 41 different API or API combinations. The participants were asked to focus on inspections of medicines containing seven targeted API: ofloxacin (OFLO), sulfamethoxazole-trimethoprim (SMTM), azithromycin (AZITH), amoxicillin-cla-vulanic acid (ACA), artemether-lumefantrine (AL), artesunate (ART) (intravenous/intra-muscular formulation) and dihydroartemisinin-piperaquine (DHAP). Genuine medicines

**Table 1. Main characteristics of the devices included in the study\*.**

| Device name | Manufacturer or Institution | Market status | Technology Main Specifications | Handheld | Cost[a] |
|---|---|---|---|---|---|
| 4500a FTIR Single Reflection Spectrometer | Agilent Technologies [9] | M | FTIR-MIR Spectral range 4,000cm$^{-1}$-650cm$^{-1}$ | N | US$ 31,067 |
| Minilab[b] | Global Pharma Health Fund E.V. [10] | M | TLC, disintegration test | N | US$ 2,510 (without reference standards) |
| MicroPHAZIR RX analyser | ThermoFisher Scientific [11] | M | NIR–Dispersive Wavelength range 1,600nm-2,400nm | Y | US$47,500 |
| NIR-S-G1 Spectrometer | Young Green Energy -Innospectra[c] [12,13] (Global Good Fund developed the smartphone application) | M[d] | NIR–Dispersive Wavelength range 900nm-1,700nm | Y | US$1,199 (without smartphone) |
| Paper Analytical Device | University of Notre-Dame [14] and Veripad [15] (Kenya, New-York and Boston) | D | Paper-based colour test | Y (S) | US$3 |
| Progeny Spectrometer | Rigaku [16] | M | Raman 1,064 nm laser | Y | (ex-demo model) |
| TruScan RM Spectrometer | ThermoFisher Scientific [17] | M | Raman 785 nm laser | Y | US$ 62,500 (including chemometric software package and tablet holder) |

\*Rapid Diagnostic Tests (RDT) and single-use immunoassay devices were deemed field-suitable in the laboratory evaluation work [7], but could not be evaluated in the present study because the developers of the single-use immunoassay test were unable to supply sufficient samples of the devices within the timeframe of the project.

D, Under development; FTIR, Fourier Transform Infrared; M, Marketed; MIR, Mid-Infrared; MS, Mass spectrometry; N, No; NIR, Near infrared; S, Single-use device; TLC, Thin-layer chromatography; Y, Yes.

a The costs reported here do not include VAT that may vary by country of purchase. Ordering several devices from the manufacturer is subject to potential reduced purchase cost.

b Unlike other devices, the Minilab was evaluated by laboratory technicians involved in current routine quality control at the National Center for Food and Drug Analysis.

c At the time of the study the NIR unit was produced by Young Green energy. It is now produced by InnoSpectra Corporation.

d The near-infrared sampling unit is marketed, but the smartphone application is not.

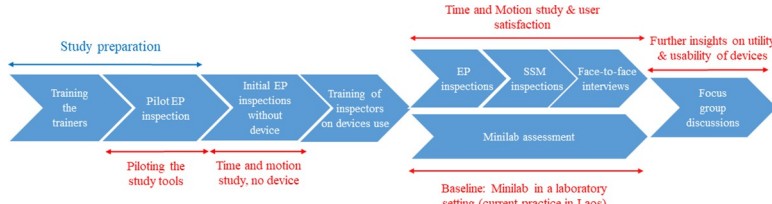

**Fig 1. Outline of the field evaluation study.** EP: evaluation pharmacy; SSM: Sample set of medicines.

were obtained from manufacturers and distributors in Laos and Thailand. Falsified versions of the antimalarial Coartem, containing none of the stated artemether or lumefantrine API, were provided by collaborators. Ultra-performance liquid chromatography (UPLC) was used as the reference technique to determine the amount of API(s) contained in FCM (**S1 Text** and **S1 Table**), except the falsified field-collected samples that had been tested by mass spectrometry [20].

## Device settings

Qualitative results obtained with the devices were based upon pattern comparison between a known good quality medicine reference and the test medicine sample data. PAD chemically reacted with the medicine ingredients generating a color pattern that was then visually compared to a reference photograph. The spectrometers computationally compared experimentally-collected spectra to reference spectra of good quality medicines stored in the device's database. Reference spectra were created for each brand tested in the study and each 'good quality' simulated medicine during the laboratory evaluation phase of the project [7]. The protocol for creation of reference spectra is available in **S2 Text**. Each sample spectrum acquired was given a score by the device software resulting from the comparison with the good quality reference library entry. Such scores needed to meet a given threshold to determine if a medicine passed. For the NIR-S-G1 spectrometer, reference samples were sent to the developer who prepared the reference libraries. The passing threshold values for the correlation coefficient or p-value testing initially set as default by the developer in the MicroPHAZIR RX, NIR-S-G1, Progeny, and TruScan RM spectrometers were utilized. These devices readout would directly tell the user 'pass' or 'fail', which were recorded. The pass threshold for the 4500a MIR spectrometer's correlation coefficient was set by us at >0.9 because the device would not output a direct pass/fail result, but rather give a list of matches with their associated correlation coefficients.

## Training medicine inspectors

Sixteen medicine inspectors, employees of the Vientiane Central Bureau for Food and Drug Inspection based in Vientiane Capital (n = 10) and Vientiane districts BFDI offices (n = 6), volunteered for the study. They were randomised to receive either a 'rudimentary' verbal training with opportunity to rehearse use of the device on a few practice samples just prior the EP inspection (5–10 min); or an 'intensive' training including verbal presentation and substantial practice with the device (1–2 h), plus an additional rudimentary verbal training and practice just prior to the EP inspection. The trainings were given by Lao post-graduate research pharmacists, trained by the lead chemist overseeing the laboratory evaluation phase (**S3 Text**).

## Evaluation pharmacy inspections

To refine the study protocol, EP inspections without devices were piloted by three pharmacy students from the Faculty of Pharmacy (University of Health Sciences, Vientiane) prior to the initial visual inspections.

Subsequently, four EP inspections by four different medicine inspectors were conducted per device, between September and December 2017. Each medicine inspector performed one simulated inspection without any device as a baseline ('Initial visual inspection'), and up to three simulated EP medicine inspections with a device. The inspectors were randomly assigned to a combination of training and devices using the Excel RandBetweenrandom number generator. Constraints were that no inspector would test more than one handheld spectrometer (Progeny, MicroPHAZIR RX or Truscan RM) due to operating procedure similarity. Only inspectors from the district office would test the NIR-S-G1 because some Vientiane BFDI office inspectors had already used the NIR-S-G1 in a previous study by our research group.

All inspections were carried out independently by the participant working alone with one device per inspection. Inspectors were asked to test any suspicious samples containing the targeted API assuming: no time limit (hence the collectors were free to inspect as many samples as they wished to), no budget restriction, that it was June 2015 (to avoid bias because some of the medicines in the EP had already expired by the evaluation) and that all blisters had no tablets missing (as some tablets were removed for analysis). They were encouraged to test samples through the blisters where appropriate. However, if they wished to perform testing requiring opening of primary packaging, the observer provided inspectors with already unpackaged samples (in a small zip-lock bag) of the same batch number of the product that were stored at the same conditions. If the inspector regarded all medicines as not suspicious at the end of the inspection, the inspector was asked to select a sample of 10% of those which did not look suspicious or passed the device test, for the Minilab testing. To reduce recall bias by medicine inspectors inspecting the EP several times, brands were changed between inspections and moved to different places.

## Sample set inspections

After each EP inspection, a pre-determined 'sample set' of medicines (SSM) was tested by each inspector with the device in an office outside the EP. These SSMs tests facilitated direct comparisons between the devices for the time taken to test a single sample and to observe user errors. The samples consisted of FCM and 'simulated' samples made during the laboratory evaluation [7] that were presented as single tablets, with packaging removed, in transparent zip-lock plastic bags labelled with the brand name, manufacturer, and dosage (S4 Text). Three of six SSM samples of either AL, SMTM or OFLO were prepared to ensure that no inspector assessed each SSM more than once. However, eight FCM samples used for the creation of five spectrometer reference library entries and one used as a test sample (an artemether-lumefantrine sample), were subsequently found to be out-of-specification by UPLC analyses, rendering these reference libraries unreliable [7]. We thus discarded spectrometer results of testing of five samples included in SSM inspections (Table 2), and eight samples included in the EP inspections. No brands were discarded from the analysis for the PAD as they use reference colour codes pictures provided by the device developers.

## Baseline screening method: Minilab

Three laboratory technicians from the National Center for Food and Drug Analysis (NCFDA) of the Lao Ministry of Health (formally trained with the Minilab and involved in training of provincial inspectors), tested the samples selected as suspicious by medicine inspectors during the 16 initial visual inspections (without device), a random set of 10% of the samples considered

**Table 2. Samples sets of medicines initially included in sample set testing.**

| API | Study Code | Brand name | Quality type and origin of the medicines |
|---|---|---|---|
| SMTM | *SPS20* | *Sulfatrim* | *G—Field-collected* |
| | *SPS21* | *Sulfatrim* | *G—Field-collected* |
| | SPS16 | Diabeta 250 | F—Look-alike[¥] - Field-collected |
| | SPS03 | N/A* | 100% API simulated medicine |
| | SPS04 | N/A* | 50% API simulated medicine |
| | SPS02 | N/A* | 0% API simulated medicine |
| AL | *SPS06* | *IPCA* | *G—Field-collected* |
| | *SPS07* | *IPCA* | *F—Field-collected* |
| | *SPS22* | *Coartem* | *G—field-collected* |
| | SPS09 | Coartem | G—Field-collected |
| | SPS10 | Coartem | F—Field-collected |
| | SPS11 | Coartem | F—field collected |
| OFLO | SPS14 | Oflocee | G—Field-collected |
| | SPS15 | Ofloxacin | G—Field-collected |
| | SPS13 | Di-Flo | G- Field-collected |
| | SPS05 | N/A* | 100% API—Simulated medicine |
| | SPS01 | N/A* | 50% API simulated medicine |
| | SPS02 | N/A* | 0% API simulated medicine |

The test sample (SPS22) and the reference library samples (n = 4: SPS20, SPS21, SPS06, SPS07) that were subsequently discarded from the results because of unexpected out-of-specifications API content as per UPLC analysis, are given in italics and highlighted. The samples with out-of-specifications reference library samples were still used for the PAD evaluation as the PAD reference libraries are independent reference pictures provided by the device developers.

G: genuine; F: falsified.

*Simulated sample

[¥] 'look-alike' medicines are defined as medicines stated as containing specific API (not one of the seven API included in this study) but the tablets were visually indistinguishable from genuine medicines in order to mimic a falsified medicine with a wrong API; the actual medicine was Diabeta (chlorpropamide), but the tablets looked identical to Sulfatrim (SMTM)] [21].

good quality during the 16 initial visual inspections, and the medicines of the three SSM, in line with the current use of the Minilab in Laos. Each technician was assigned to the testing of all samples of two or three API (e.g. inspector A tested all the samples of SMTM and AL).

## User satisfaction and focus group discussions

After completion of each SSM testing, the medicine inspectors were asked five open-ended questions, through face-to-face interviews in Lao language (S2 Table). Two months after the last EP inspection, three focus group discussions (FGD) were held. Each FGD had five medicine inspectors to give further insight into the utility and usability of the tested devices to support PMS systems (S3 Table).

## Outcomes

In the absence of spectrometer manufacturer's guidelines, when a sample failed the first test with a device, medicine inspectors were instructed to operate a 'best of three' system for overall sample classification. Three tests were performed with the device on the failing samples, the most frequently occurring of 'pass' or 'fail' would then be the overall sample classification. For

the PAD, inspectors were instructed to re-run failing samples once, as recommended by the developer. If the sample failed again, the sample was classified as failing. For both the EP and the SSM inspections, medicine inspectors were asked to record the sample identifier, pass/fail results of each single test, and the overall pass/fail classification on a recording sheet (**S4 Table**). Data analysis was performed using results from the inspector's overall pass/fail classification of the sample.

Time and motion studies were conducted by two observers (only one observer for the Mini-lab). In EP inspections, one observer unobtrusively, with no conversation allowed with the participant, recorded the times to perform specific tasks on a recording sheet (**S5 Table**). Another observer recorded deviations from device protocol ('user errors'). For SSM, two observers recorded the times to perform specific tasks. The tasks recorded in EP were the times spent to conduct sample visual inspections, testing with the device ('sample testing'), and interpret/record the results (**S6 Table**). For SSM no visual inspection was conducted by the inspectors, as the tablets were provided outside their packaging. In addition to the time to interpret/record the results, two phases were identified as part of the SSM 'sample testing'; 'sampling' (started when use of the device or removal of the tablet from the packaging to begin testing started; ended when the process to obtain a result is started) and 'device testing' (started when the process to obtain a result is started; ended when result is obtained).

## Data analysis

The median and interquartile range (IQR) number of samples wrongly classified, and the percentage with 95% confidence intervals (CI) of samples wrongly classified over all the EP inspections per device, are presented. Fisher's exact tests were used for the comparisons of the proportions of the number of samples wrongly classified by device pairs. Wilcoxon rank-sum tests were used to compare the number of samples wrongly categorised with and without devices in EP inspections. For SSM differences in accuracy in correctly classifying samples between devices were examined using mixed effects logistic regression yielding adjusted odds ratios, adjusted for training type (rudimentary/intensive), and sample set type as factors and inspectors as cluster-specific random effects.

The total time spent in EP inspection, time spent per phase during SSM, and total time spent per sample in SSM testing are described using medians (IQR). Wilcoxon rank-sum tests were used to test the differences in the times between the initial EP visual inspection and EP inspection for each device. For SSM testing, differences in the times between devices were examined using mixed effect generalised linear regression models to obtain the estimated devices' effect compared to the reference devices, adjusted for training group and sample set as factors and inspectors and observers as cluster specific random effects. The data demonstrated skewed distributions for time and we therefore used the variable transformed to natural logarithm.

All tests were performed using a 5% (0.05) significance level. Microsoft Excel 2013 and STATA version 14.0 were used for analyses.

The user error(s) observed during EP and SSM inspections are summarized as narratives by category of errors (e.g. selection of the wrong reference library for spectrometers). The information of face-to-face interviews and the FGD are summarized and presented as narratives highlighting emerging common themes. More details by device are provided in **S5 Text**.

## Results

### Times results

**Time to inspect the evaluation pharmacy.** EP inspections with each device took significantly longer to complete compared with the initial visual inspections (25 min 16 s) without

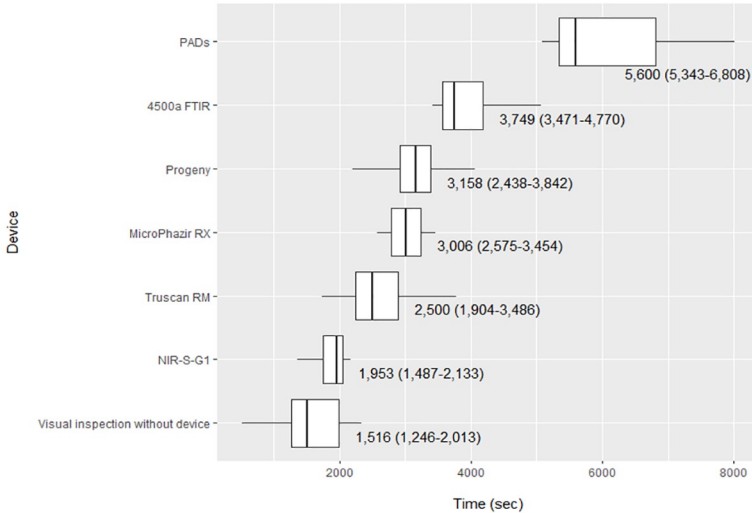

**Fig 2. Time spent inspecting evaluation pharmacy, by device.** Values in the figure are medians (IQR); 2,000 seconds is ~ 33 minutes and 8,000 seconds is about 2 hours.

devices (p<0.05, Wilcoxon rank sum), except for the NIR-S-G1 (32 min 33 s, p = 0.307) (**Fig 2 and S7 Table**). Visual inspection duration when using a device was significantly shorter for all devices than for initial visual inspections of the EP with no device, except for the 4500a FTIR (p = 0.061). During more than one-third of the inspections with devices (n = 9, 41%), inspectors spent less than one minute in sample visual inspection (**S7 Table**). As one inspector did not perform the negative control of the PAD and the observers failed to record the calibration time with the MicroPHAZIR RX during one inspection, these data were excluded from the analyses.

**Time per sample in sample set inspections.** For SSM inspections, the median time to test one sample ranged from 94 sec for the NIR-S-G1 to 2,063 sec (34 min 23 s) for the Minilab. The Minilab and PAD took significantly longer total times per sample compared to other devices (p<0.001)(**Table 3**).

The NIR-S-G1 had a significantly shorter total time per sample (median 94 s) than any other devices tested (p<0.001); sampling was significantly faster than for the other devices and in interpreting/recording compared to all the devices except the MicroPHAZIR RX (median of 14 s vs 22 s, p = 0.78) (**Table A in S8 Table**). The MicroPHAZIR RX was significantly faster

**Table 3. Pairwise comparisons of the median total time taken per sample in sample set testing.**

|  | 4500a FTIR | MicroPHAZIR RX | Minilab | NIR-S-G1 | PAD | Progeny | Truscan RM |
|---|---|---|---|---|---|---|---|
| **4500a FTIR** | - | <0.001 | <0.001 | <0.001 | <0.001 | 0.004 | 0.009 |
| **MicroPHAZIR RX** | - | - | <0.001 | <0.001 | <0.001 | <0.001 | 0.002 |
| **Minilab** | - | - | - | <0.001 | <0.001 | <0.001 | <0.001 |
| **NIR-S-G1** | - | - | - | - | <0.001 | <0.001 | <0.001 |
| **PAD** | - | - | - | - | - | <0.001 | <0.001 |
| **Progeny** | - | - | - | - | - | - | 0.51 |
| **Truscan RM** | - | - | - | - | - | - | - |
| **Median total time per sample (IQR)/secs** | 316 (206–373) | 134 (98–170) | 2,063 (1,766–2,920) | 94 (61–112) | 620 (562–716) | 273 (163–302) | 148 (109–299) |

P-values of the mixed effects generalised linear regression model of ln(total time) adjusted by device and training, and clustered by inspectors and observers

**Table 4. Pairwise comparisons of the percentage of samples wrongly classified over all inspections out of total samples tested overall with the devices in the evaluation pharmacy inspections.**

| | 4500a FTIR | MicroPHAZIR RX | NIR-S-G1 | PAD | Progeny[b] | Truscan RM[b] |
|---|---|---|---|---|---|---|
| **4500a FTIR** | - | 0.103 | 1.000 | 0.014* | 1.000 | 0.242 |
| **MicroPHAZIR RX** | - | - | 0.243 | <0.001*** | 0.167 | N/A[a] |
| **NIR-S-G1** | - | - | - | 0.005** | 1.000 | 0.269 |
| **PAD** | - | - | - | - | 0.023* | <0.001*** |
| **Progeny** | - | - | - | - | - | 0.225 |
| **Truscan RM** | - | - | - | - | - | - |
| **% samples wrongly classified (95% CI)** | 9.7 (2.0–25.8) | 0 (0–10.3) | 7.7 (1.6–20.9) | 37.9 (20.7–57.7) | 8.3 (1.0–27.0) | 0 (0–13.2) |

P-values of the Fisher's exact test are presented

* p<0.05

**p<0.01

***p<0.001

a Not applicable as no samples were wrongly categorised in inspections with the Truscan RM or MicroPHAZIR RX

b Artesunate samples were discarded from the results analysis because samples were scanned through the glass vials by the inspectors, although reference library was created by scanning through a replacement packaging (plastic packaging)

in testing one sample than all other devices except the NIR-S-G1. The Progeny was significantly slower in device testing and interpreting/recording times per sample than the MicroPHAZIR RX and the Truscan RM (p<0.001). The PAD and 4500a FTIR sampling times were not significantly different (4 min 2 s and 3 min 49 s, respectively, p = 0.059).

The inspectors with rudimentary training did not spend longer testing one sample, compared to the inspectors with intensive training, adjusted for devices, sample set tested, and clustered by inspectors and observers (p = 0.11, **Table B in S8 Table**).

## Device accuracy

Over all EP inspections, samples were wrongly categorized with a frequency of 0% with the Truscan RM and MicroPHAZIR RX, to 37.9% (95% CI, 20.7–57.7%) with the PAD (**Table 4**). Significantly more samples were wrongly classified with the PAD compared to all other devices (p<0.05). All incorrect classification results were for genuine medicines being classified as suspicious (false positive).

The median numbers of samples wrongly classified in EP inspections with the 4500a FTIR [1 (0.3–1)], MicroPHAZIR RX [0 (0–0)], NIR-S-G1 [1 (0.3–1)] and Truscan RM [0 (0–0)] were significantly lower than in initial visual inspections [p = 0.048, p = 0.008, p = 0.048 and p = 0.005, respectively] (**Table 5**). There were no statistical differences in the number of samples wrongly classified in EP inspections with the PAD [2 (1–5.3)] and Progeny [0 (0–1.5)], compared to initial visual inspections (p = 0.631 and p = 0.059, respectively).

For SSM inspections there were no significant differences between devices that wrongly classified samples as suspicious or not suspicious, adjusted by training status, sample set tested, and clustered by inspectors (**Table 6**).

Over all SSM inspections, 10 out of 18 (55.6%) misclassifications were false negative results with samples containing 50% of OFLO or SMTM, two (11.1%) were false negative falsified FCM stated to contain AL and six (33.3%) were false positive samples of OFLO and SMTM.

The two 50% API samples tested with both the MicroPHAZIR RX and NIR-S-G1 were correctly classified as suspicious whereas the 4500a FTIR correctly classified 1/2, the Minilab 0/2, the PAD 3/4, the Progeny 0/3, and the Truscan RM 1/4.

**Table 5. Comparison of the number of samples incorrectly classified in evaluation pharmacy inspections with devices vs initial visual inspection without device.**

| Device | Z | p-value | Median (IQR) number of samples wrongly classified with the device | Median (IQR) number of samples wrongly classified in initial inspection$ |
|---|---|---|---|---|
| 4500a FTIR | -1.980 | 0.048* | 1.0 (0.3–1.0) | 2.0 (1.0–2.3) |
| MicroPHAZIR RX | 2.638 | 0.008** | 0 (0–0) | 2.0 (1.0–2.3) |
| NIR-S-G1 | 1.980 | 0.048* | 1.0 (0.3–1.0) | 2.0 (1.0–2.3) |
| PAD | -0.480 | 0.631 | 2.0 (1.0–5.3) | 2.0 (0.8–2.3) |
| Progeny | 1.891 | 0.059 | 0 (0–1.5) | 1.5 (1.0–2.3) |
| Truscan RM | 2.814 | 0.005** | 0 (0–0) | 1.5 (1.0–2.3) |

Z statistic and p-value of the Wilcoxon rank sum test are presented

* $p < 0.05$

** $p < 0.01$

*** $p < 0.001$

$ The numbers of samples wrongly categorized in initial inspections without devices used in the comparisons vary because we included only brands tested in initial inspections that each device were able to test (e.g. AL samples wrongly categorized during initial inspections were excluded for the PAD, as the PAD could not test samples containing AL). In both initial inspections without devices and inspections with devices, we excluded samples wrongly categorized from brands subsequently found to have reference library spectra obtained from poor quality reference samples (as per UPLC analyses), except for the PAD.

Inspectors with rudimentary training were not significantly more likely to wrongly classify the samples compared to those with intensive training in SSM [OR 1.5 (95% CI 0.5–4.9)] adjusted by devices and sample set tested and clustered by inspectors (**S9 Table**).

## Observed user errors

The main observed user errors with the MicroPHAZIR RX, NIR-S-G1, Progeny, and Truscan RM were the selection of wrong comparator reference libraries in 3.9%, 27.0%, 7.5% and 20.0% of the scans in EP inspections and 0.0%, 27.8%, 0.0% and 25.8% in SSM inspections, respectively (**Table 7**). The inspectors recognized all their errors and repeated the tests during MicroPHAZIR RX and Progeny inspections, resulting in no overall misclassification of samples. However, in 16 of 17 scans (88.9%) using wrong reference libraries with the Truscan RM and 21/27 (77.8%) using the NIR-S-G1, the users did not realize the errors. In all these cases, the wrong brand (a different brand stated to contain the same API and strength) was selected by the users as the reference library. None of the 11 samples scanned with wrong reference

**Table 6. Matrix of pairwise comparisons of accuracy of devices in classifying samples incorrectly during sample set inspections (test device in row vs reference device in column).**

|  | 4500a FTIR | MicroPHAZIR RX | Minilab | NIR-S-G1 | PAD | Progeny | Truscan RM |
|---|---|---|---|---|---|---|---|
| 4500a FTIR | - | 1.2 (0.1–25.0) | 0.5 (0.0–7.5) | 0.5 (0.0–5.8) | 0.5 (0.0–4.9) | 0.3 (0.0–3.5) | 0.6 (0.1–7.5) |
| MicroPHAZIR RX | - | - | 0.4 (0.0–6.5) | 0.4 (0.0–5.3) | 0.4 (0.0–3.8) | 0.2 (0.0–2.8) | 0.5 (0.0–6.1) |
| Minilab | - | - | - | 0.9 (0.1–10.2) | 0.9 (0.1–6.5) | 0.6 (0.1–4.2) | 1.3 (0.2–10.8) |
| NIR-S-G1 | - | - | - | - | 1.0 (0.1–6.9) | 0.6 (0.1–3.0) | 1.4 (0.2–10.9) |
| PAD | - | - | - | - | - | 0.4 (0.1–2.5) | 1.4 (0.3–7.0) |
| Progeny | - | - | - | - | - | - | 2.3 (0.4–13.0) |
| % samples wrongly classified (95%CI)$ | 5.6 (0.1–27.3) | 7.7 (0.2–36.0) | 11.8 (1.5–36.4) | 11.1 (1.4–34.7) | 20.8 (7.1–42.2) | 23.5 (6.8–49.9) | 15.0 (3.2–37.9) |

Odds ratio (95% CI) of the mixed effect logit model, with adjustment on the type of training received (rudimentary or intensive), sample set type (OFLO, AL, SMTM), and clustered by inspectors, are presented

$ 95%CI for binomial distribution

**Table 7. Observed user errors during EP and SSM inspections.**

| Device | Selection of wrong reference libraries in EP | | | Selection of wrong reference libraries in SSM | | | Other errors | | |
|---|---|---|---|---|---|---|---|---|---|
| | Scans % (n/N) | Samples % (n/N) | % (n/N) of samples misclassified (N = total number tested) | Scans % (n/N) | Samples % (n/N) | % (n/N) of samples misclassified (N = total number tested) | Description | % (n/N) of samples misclassified (N = total number tested) | Comments |
| **4500a FTIR** | N/A | N/A | N/A | N/A | N/A | N/A | 5.9% (3/51) scans in EP and 3.0% (1/33) in SSM were not renamed after acquisition in the device memory | 0 | Samples were recorded on paper by the inspectors. Thus errors did not result in sample misclassification, but could affect traceability in practice |
| **MicroPHAZIR RX**[*] | 3.9% (2/51) | 5.9% (2/34) | 0.0% (0/34) | 0.0% (0/33) | 0.0% (0/13) | 0.0% (0/13)[$] | 5.9% (3/51 scans with tablets not inserted in sample cover) in EP | 0[$] | All errors made by inspector with rudimentary training |
| **NIR-S-G1** | 27.0% (17/63) | 28.2% (11/39) | 5.1% (2/39) | 27.8% (10/36) | 33.3% (6/18) | 11.1% (2/18) | None | None | |
| **PAD** | N/A | N/A | N/A | N/A | N/A | N/A | Reading result errors in 24.1% (7/29) samples tested in EP and 20.8% (5/24) in SSM | 6.9% (2/29) in EP and 16.7% (4/24) in SSM | In some cases both the PAD showed wrong colours and the user made an error of interpretation, leading to overall correct classification (more details in S5 Text. Results of the evaluation by device) |
| | | | | | | | None of the failing samples were rerun despite clear instructions to rerun suspicious samples | Uncertain | |
| | | | | | | | Use of the same visibly contaminated water for multiple PAD during one EP inspection (inspector with rudimentary training) | Uncertain | |
| **Progeny**[¥] | 7.5% (4/53) | 13.3% (4/30) | 0.0% (0/30)[$] | 0.0% (0/21) | 0.0% (0/13) | 0.0%[$] | Deviation from study protocol: One inspector did not run the 'Application' test after running the 'Analyse' function | 0 | |
| **Truscan RM** | 20.0% (9/45) | 19.4% (6/31) | 0.0% (0/31) | 25.8% (8/31) | 20.0% (5/20) | 0.0% | None | None | Inspectors did not recognize they selected the wrong library entry, but the device returned correct result |

[*]3 inspections only with the MicroPHAZIR RX because results of one inspection were discarded because of an issue over the inbuilt reference library

[$]Errors were recognized by the inspectors who re-tested the samples without mistakes

[¥] Wrong selection of the reference library can happen only with the 'Application' function. Using the analyse function, no user errors were observed during both EP and SSM

libraries for which the users did not recognize the errors were misclassified with the Truscan RM. However, four samples out of the 17 samples tested with wrong reference library with the NIR-S-G1, for which the users did not recognize the errors, were incorrectly classified (all were false positives). Two out of 29 samples tested (6.9%), and four out of 24 samples tested (16.7%) were misclassified as a result of errors in PAD interpretation in EP and SSM inspections, respectively.

### User satisfaction

All spectrometers, except the NIR-S-G1, were felt to be heavy and/or rather cumbersome. The portable 4500a FTIR spectrometer was perceived as suitable in inspections of manufacturing and distributing sites by inspectors who liked the extra information given by the table of results (table of matches, with its list of API and % match) as this was felt to increase confidence in the device results. However, the 4500a was identified as not suitable for routine pharmacy inspection due to its large size and the need for sample crushing and for cleaning the sampling window: *"[. . .] in most of the big pharmacies in our country there's no space to test, people queue for hours to get their medicines; there's no way to place the heavy device like this and computer and if we want to test it's just rarely possible."*

The MicroPHAZIR RX was described as easy to use, reliable, comfortable, and fast. The sample window indicator, which shows the inspector whether the sample is sufficiently covered by the sample window to produce a reliable result, was cited as a helpful additional feature giving inspectors additional confidence in their sampling technique. The device froze during one of the four EP inspection and all the records were lost, which made that inspector think that the use of the device would be '*wasting of time*'.

The NIR-S-G1 was singled out by medicine inspectors as well-suited for any level of the supply chain due to its small size, fast testing time, and easy-to-use smartphone application. However, the lack of capability to create and update the reference library of comparators locally was perceived as a key limitation.

Although medicine inspectors liked the PAD lack of reliance on electricity or sophisticated instrumentation, the need to prepare samples, to have a working space to carry out the analysis, and the longer experiment time were frequently raised as concerns with regards to their usability in pharmacies and at distributors sites. Difficulties in interpreting the results were often highlighted: *"[. . .] for example in the protocol it's said it's pink and in reality it's a faded pink so it depends on the user's eyesight and his/her decision. So, it's difficult to tell the actual color."* Two medicine inspectors acknowledged that the PAD would be useful to test raw materials at manufacturers.

Medicine inspectors liked the ability of the Progeny to display more than a 'pass' or 'fail' result. It was felt to be quite slow to scan, and three inspectors commented that the touchscreen was not very responsive. Three inspectors stated that the supplied tablet holder was difficult to use with small tablets. It was noted as interesting for inspections in manufacturers, distributors, or border points, but of limited use in pharmacy outlets.

The Truscan RM was perceived as easy and comfortable to use, but with a slow device testing time (this was raised during face-to-face interviews). It was deemed more suitable for use in inspection of manufacturers' plants, distributors, or border check points rather than in pharmacy outlets.

### Discussion

This pilot study provides insight into the performances, the advantages/limitations of six screening devices in the hands of Lao medicines inspectors in simulated inspections. The

NIR-S-G1 was the fastest spectrometer to test one sample whilst the PAD and the Minilab took significantly longer to test one sample than all the spectrometers. Within the limited context of this study, the five spectrometers showed promising accuracies to identify falsified and genuine medicines. Inspectors' difficulties to read and interpret colour barcodes with the single-use PAD may have led to a lower accuracy of these compared to other devices in the evaluation pharmacy. The selection of the wrong reference libraries by inspectors was observed with all spectrometers but these errors did not lead to final erroneous classifications of medicines, except in four cases with the NIR-S-G1. Findings also suggest that policy makers wishing to implement devices in PMS should be aware that overconfidence in devices risks harm by reducing inspectors' visual inspection investment.

Of the six devices studied, five were spectrometers and there were no significant differences in the performance outcome measured in EP or SSM inspections between them in our limited data set. The inspections of the EP with all spectrometers except the Progeny were more accurate than the initial visual inspection with no device. Most spectrometers, except for the NIR-S-G1, were felt too heavy and cumbersome for pharmacy inspections. In one inspection, the MicroPHAZIR RX was used with the device resting on the bench of the simulated pharmacy rather than as a 'handheld' device. The handheld NIR-S-G1 was perceived as light, handy and user-friendly, thus suitable for routine inspections at any level of the supply chain.

The Truscan RM, Progeny (using the 'application' function), MicroPHAZIR RX and the NIR-S-G1 all gave a simple 'pass/fail' result, a feature appreciated by the medicine inspectors. The 'matching' values given by the Progeny (using the 'analyse' function) and the 4500a FTIR gave reassurance in the results. Except the 4500a FTIR, the spectrometers require the user to select the correct reference library entry for comparison with the tested spectrum. Out of the samples tested with the NIR-S-G1 with the inspector selecting the wrong reference library, almost one-fourth gave false classifications. In instances where the wrong reference libraries were selected with the TruScan RM, the Progeny, or the MicroPHAZIR RX, overall classification of the sample as suspicious or not suspicious by the inspector were not compromised. Indeed, with the MicroPHAZIR RX and Progeny, the mistakes were all recognised by the inspectors who repeated the analysis. In all cases with the Truscan RM, the library selection errors were not recognized by the inspectors but the device gave the correct result and the samples were accurately classified overall. There appeared to be a lack of awareness that different brands of the same API may contain different excipients, resulting in need for different reference libraries. Indeed, in all the cases the wrong 'brands' library entries selected by the inspectors were that of medicines containing the same API(s) at the same strength than the tested medicine, such as selecting Sulfatrim instead of Vactrim (both containing SMTM). In some instances, the result shown by the device was that expected if the correct library had been selected. For example, in the case of Sulfatrim vs. Vactrim, the device gave a 'pass' to the Vactrim tablet being tested against the Sulfatrim reference library. It is likely that in some cases different brands containing similar API and excipient compositions lead to a correct overall classification of the samples because the medicines' are chemically similar. Although little is known about the variability of response of portable spectrometers to different brands of the same API(s), our findings suggest that the Raman devices may be less susceptible to formulation-specific signature variations than the NIR [22]. Improvement of the function to select the reference library in the NIR-S-G1 and using the in-built barcode reader (featured in the Truscan RM, MicroPHAZIR RX, and Progeny) are likely to reduce the risk of wrong library selection and thus the number of incorrect results.

The MicroPHAZIR RX and the NIR-S-G1 correctly classified all 50% API medicines tested, whereas the other spectrometers correctly classified none to less than half of them. Identification of substandard samples containing between 70 and 90% of API with spectrometers, and

those containing higher than the upper limit of the specifications-commonly found in field surveys [23,24], should be further explored.

Whilst the spectrometers were expected to give information on the dosage formulation of the tested samples, the PADs were designed to indicate whether specific API(s) or excipients were present in the tested samples, but not to identify samples containing lower than stated amount of the expected API. Hence, caution is needed to interpret the results of their comparison with the spectrometers. The PADs required the user to make subjective judgement on the visual likeness of the test sample result to the reference result. This is likely to have contributed to the significantly higher number of samples wrongly categorised by the medicine inspectors as compared to other devices in the EP inspections. Problems with colour interpretation were also observed for the Artemisinin derivative test (ADT), despite the high level of confidence in the results expressed by the laboratory technicians who were newly trained to use the ADT [25]. These and issues of colour blindness are likely to be greatly helped by automated smartphone interpretation software with image analysis software, such as ImageJ [26–28]. In addition, although the inspectors were told to change the water used for the PAD before running a new sample if water contamination occurred, one inspector, with rudimentary training, did not. This supports the impression that the training given may have been insufficient. For the advantages of the PAD to be realised, training schemes with user proficiency testing, continuing education, certification and quality control will be necessary, as they will be for the spectrophotometers. Interestingly, although they are not designed to identify samples with API content below specifications, the PAD identified three out of four 50% API samples tested. Expert readers had previously identified 19/20 samples containing 40% of the stated amount of chloroquine as giving a 'weak' signal, compared to the 18/18 full-strength formulations giving a 'strong' signal, but mixed results were observed in identification of samples with 70% stated API [29]. The PAD deserve more consideration as devices for semi-quantitative detection of substandard antimicrobials that are probably important drivers of antimicrobial resistance. Indeed, samples containing low ceftriaxone concentrations were successfully identified with high sensitivity by a novice PAD user and ImageJ, showing promising results for identification of substandard samples containing less than 80% of the stated API [28]. Recently, the μPAD, a competitive enzyme assay on paper, has shown encouraging results in the hands of five users unfamiliar with the device, to identify falsified beta-lactams [30]. The semi-quantitative properties of the ChemoPADs, using similar features to the PAD, coupled with an image analyser identified 11 of 20 substandard cisplatin samples found in Ethiopia [27].

As expected, devices requiring sample preparation and user data interpretation (4500a FTIR, PAD, Minilab) took the inspectors significantly longer time per sample than those which do not. This was particularly pronounced for both the PAD and the Minilab, but this may be offset by their ability to run more than one sample concurrently. The NIR-S-G1 and MicroPHAZIR RX were the fastest devices to test individual samples. At the time of the study, the NIR-S-G1 did not have the ability to record sample details on the device and did not have a sample holder for unpackaged tablets. This contributed to its fast speed of analysis, but there may be inconsistency with tablets that are too small or do not fit flush against the sampling window.

The time spent on visual inspection in the EP was significantly shorter when using a device than for initial visual inspections alone, except for the 4500a FTIR (p = 0.061). The selection of suspicious samples by visual inspection may be key to identify poor quality medicines, especially those with obvious defects such as discoloration or typographic errors [31]. Therefore, reduction in visual inspection time may have negative consequences in finding suspicious medicines samples. Hence, it is possible that device introduction could be counterproductive depending on the prevalence of SF medicines that could be visually recognised. Instead of

visually inspecting different blisters of the same brand as in the inspection without device, it seems that the inspectors chose to test a sample of one packet of a brand with the device, taking that result to be representative of all samples from that brand. This may be an artefact of the experimental set-up as the EP was inspected without devices first. When questioned about why they chose not to do visual inspection, some inspectors replied that they would expect samples of the same brand of medicine in the same pharmacy to be from the same batch, hence of identical quality. The paucity of visual inspection of samples could also be related to the increased perceived time pressure to complete an extra task within the 'normal' pharmacy inspection time. Further work is needed to investigate these findings and the impact of device use on real-life inspection effectiveness.

Non-destructive testing of samples is preferable for pharmacy inspections [6]. The lack of budget to buy medicines to test, and the waste of samples for the pharmacy being inspected were mentioned by medicine inspectors as pitfalls of destructive technologies. Even for non-destructive devices, testing can currently only be carried out through transparent packaging. Of the fourteen brands of the target EP medicines, ten were in opaque packaging and therefore had to be removed from the packaging (thus 'destroyed') prior to testing. Innovations to blister pack and packaging could facilitate accurate spectroscopic evaluation [4]. We have been unable to find discussion as to the impact of different plastics on spectral acquisition. Spatially offset Raman spectroscopy (SORS) technology deserves investigation to scan through opaque packaging [32,33].

Few have discussed training requirements for device users [34–39], and limited scientific evidence can be retrieved from studies that were not primarily designed for that purpose. All the inspectors in our study were able to successfully complete pharmacy inspection and sample set testing with the devices regardless of the training they had received. In our limited data set, inspectors with intensive training were not significantly more likely to correctly classify the samples as suspicious or not than those with less training.

Important limitations exist in our study and those, as well as the difficulties to perform such research encountered in our study are listed in **Box 1**.

## Box 1. Limitations and difficulties encountered to perform this research study

- For the spectrometers, only one unit of each device was evaluated. We therefore make no assessment of variability between different units of the same device type.

- Only six API/combinations of APIs, all antimicrobials, and all sourced from one region were evaluated. One API (DHAP) of the seven initially selected for investigation had to be removed from analysis due to poor quality samples being used in construction of the device reference libraries.

- Only one parenteral formulation was investigated; all other samples were formulated as tablets. No testing of topical/liquid/capsule dosage forms was conducted.

- For laboratory-created spectrometer reference libraries (does not include NIR-S-G1, for which the developer created the reference library):

  ○ Manufacturer-set default values were used with no attempt to optimise these for specific medicines tested

○ Limited consideration of batch-to-batch variability for field-collected medicines

- Evaluation pharmacy included a small proportion of falsified medicines (3/~110 blisters stocked)

- Due to limited stock, some samples had exceeded their expiry date. Inspectors were specifically asked to overlook important normal cues for visual inspection (expiry date, inclusion on national list of registered medicines, condition of packaging, storage conditions). Overlooking these cues during inspection of the evaluation pharmacy has limited resemblance to the inspectors' standard practice.

- The field-study team did not receive any direct training from the manufacturer and followed protocols in a second language.

- The 4500a FTIR does not give a pass/fail result and requires interpretation. Bias may have been introduced in the measure and comparisons of effectiveness between the 4500a FTIR and other devices because of instructions given to the inspectors that were incorrect, due to a misunderstanding by the trainer (**S5 Text**).

- Whenever possible, the EP stock consisted of complete blisters, in original packaging. Medicines inspectors were encouraged to test samples through the blisters where appropriate. If needed, they were provided with already unpackaged samples. This was because of the limited number of samples available for the study in the EP, especially falsified medicines, and to preserve the complete blisters to avoid inspection bias introduced by progressively having more incomplete blisters stocked in the pharmacy.

- To avoid recall bias by medicine inspectors inspecting the EP several times, some of the brands stocked in the pharmacy were changed between inspections, samples were moved to different places, and the samples stocked in the pharmacy was thus not always consistent between inspections.

- Opinions from the inspectors were formed in the context of a 'routine' pharmacy inspection. Use of devices in different contexts, e.g. by manufacturers, or in a basic laboratory such as might be found at a provincial level, may have resulted in different user opinions.

- Samples of parenteral artesunate powder were scanned with the Truscan RM and Progeny through the glass vials by the inspectors, although the reference library was created by scanning through a replacement packaging (plastic packaging). These results were discarded from our analysis.

- We did not investigate whether the plastics of primary packaging were different between batches used for the creation of the reference libraries and the tested samples. Samples were always stored in fridges away from light. Although this was not investigated in our work, we believe that the time between the creation of the reference libraries 'through the blister' and the field study (6 months) was brief enough to minimize the risks of wrong conclusions due to degraded plastics.

- Some samples were found to be poor quality by UPLC analysis, but results were not available until after completion of the study. As a result, we did not have access to good reference library comparators, and it was decided to discard results of the 13 affected samples.

- The number of inspections carried out with each device and the number of samples stocked, particularly of SF medicines, were limited.

- The devices used in this study may have evolved since this study was performed and their performances may have changed. The results of this study may not reflect these changes.

- A statistical interaction between 'device' and 'inspector' was not included in our statistical analyses. This would have led to overparameterization of the models. We included inspectors as cluster-specific random effect.

As an independent public health investigation performed in a field setting, this exploratory study gives evidence on some aspects of the use of devices in the field, to facilitate MRAs decisions as to whether these new technologies are appropriate for screening of diverse medicines in their countries. This article is part of a series of publications describing studies of multiple aspects of the use and implementation of devices in PMS such as their costs, cost-effectiveness, and barriers identified for their implementations (e.g. the difficulties for creating quality reference libraries for spectrometers) [7, 40–42]. These should be considered when making decisions on the best devices to use in specific settings. Without further objective validation, device implementation should be cautious. Their advantages, limitations, and cost-effectiveness [40] should be clearly understood and further investigated. However, the innovation of testing devices in an evaluation pharmacy holds promise for enhancing our understanding of their use between laboratory and real-life field scrutiny. With further work such devices hold great promise to empower medicine inspectors globally.

## Supporting information

**S1 Text. UPLC confirmatory methods protocols.**
(PDF)

**S2 Text. Basic processes for qualitative spectral comparison and protocols for reference library creation. Text A. Illustration of the basic processes for qualitative spectral comparison. Fig A. Illustration of the process for reference library creation and spectral comparison analysis. Text B. The MicroPHAZIR RX spectrometer. Text C. The 4500a FTIR spectrometer. Text D. The Progeny spectrometer. Text E. The Truscan RM spectrometer. Text F. Difficulties encountered during reference library entries creation.**
(PDF)

**S3 Text. Instructing the trainers and inspectors training in the use of the devices.**
(PDF)

**S4 Text. Simulated Medicine Preparations. Text A. Details about simulated medicine preparation. Table A. Formulations of simulated medicine preparations.**
(PDF)

**S5 Text. Results of the evaluation by device. Text A. 4500a FTIR Single Reflection**. Table A. Results from evaluation pharmacy inspections with the 4500a FTIR by four inspectors. Table B. Results from sample set testing for the 4500a FTIR: AL and OFLO sample set tested twice by a total of 4 inspectors. **Text B. MicroPHAZIR RX**. Table A. Main errors made by the three inspectors during the evaluation pharmacy inspections with the MicroPHAZIR RX.

Table B. Performance of the MicroPHAZIR RX during evaluation pharmacy inspections by three inspectors. Table C. Results from sample set testing with the MicroPHAZIR RX (SMTM, OFLO, and AL sample sets, each tested once by one inspector). **Text C. Minilab**. Table A. Results from Minilab testing of sample sets conducted by 3 FDQCC Lao technicians. **Text D. NIR-S-G1 (Beta Version)**. Table A. Main errors made by four inspectors during the evaluation pharmacy inspections with the NIR-S-G1. Table B. Performance of the NIR-S-G1 during evaluation pharmacy inspections by four inspectors. Table C. Results from sample set testing with NIR-S-G1. **Text E. Paper Analytical Devices (PAD)**. Table A. Performance of the PAD during evaluation pharmacy inspections by four inspectors. Figure A. Inspector record sheet (left) for an AZITH sample (in blue pen). Lane interpretation instructions for AZITH are given (right). Table B. Main errors made by four inspectors during the evaluation pharmacy inspections with the PAD. Table C. Results from sample set testing–Paper analytical devices. **Text F. Progeny**. Table A. Number of samples tested and scans performed using analyse or application functions during four inspections of the evaluation pharmacy with the Progeny. Table B. Performance of the Progeny during evaluation pharmacy inspections by four inspectors. Table C. Results from four sample sets tests (SMTM by two inspectors, OFLO and AL by one inspector each) with the Progeny. **Text G. Truscan RM**. Table A. Results from evaluation pharmacy inspections with Truscan RM by four inspectors. Table B. Performance of the Truscan RM during evaluation pharmacy inspections by four inspectors. Table C. Results from sample set testing with the Truscan RM.
(PDF)

**S1 Table. Main characteristics and UPLC results of the medicines utilized in the evaluation.**
(PDF)

**S2 Table. User satisfaction questionnaire.**
(PDF)

**S3 Table. Outline of the focus group discussions (5 inspectors per group discussions).**
(PDF)

**S4 Table.** Extract of the inspection record sheet for (A) Evaluation Pharmacy inspections (B) Sample set of medicines inspections. Note that the record sheets were adapted for the PAD, Progeny, and 4500a FTIR which are interpreted differently than other devices (e.g. inspectors were asked to fill in which column of the PAD they read for interpreting the results). Table A. Evaluation pharmacy inspection. Table B. Sample set medicine inspections.
(PDF)

**S5 Table.** Time and motion study observer recording sheet in a) Evaluation Pharmacy inspection b) Sample Set of medicines inspection. Note that record sheets were adapted for the Paper Analytical Cards. Table A. Evaluation pharmacy inspection. Table B. Sample set inspections.
(PDF)

**S6 Table. Definition of the times measured in the evaluation pharmacy and sample set inspections.**
(PDF)

**S7 Table.** Time spent inspecting the evaluation pharmacy by phase–(A) Wilcoxon rank sum test results and (B) primary data. Table A. P-values of the Wilcoxon rank sum test (times are not normally distributed) results for the comparison between evaluation pharmacy inspection with specified device vs initial visual inspection are presented. Table B. Time spent inspecting

evaluation pharmacy by phase—primary data.
(PDF)

**S8 Table. Median (IQR) times (seconds) per sample per device in sample set testing and results of the mixed effects generalised linear regression model. Table A. Median (IQR) sampling, device testing and recording times (seconds) per sample per device in sample set testing.** Table B. Factors influencing the total time per sample [ln(total time)] in sample set testing—mixed effects generalised linear regression model (with inspectors and observers as random effects).
(PDF)

**S9 Table. Factors influencing the wrong classification of samples in sample set testing— mixed effects logistic regression (with inspectors as cluster-specific random effects).**
(PDF)

## Acknowledgments

We are very grateful to the Government of the Lao PDR for their support, especially the Bureau of Food and Drug Inspection, the Food and Drug Department, the National Center for Food and Drug Analysis and the University of Health Sciences. We are grateful to Miss Viphavanh Soulaphy, Miss Orlathai Saiyasane, Miss Thipphaphone Keonakhone, Miss Sonethalee Senboutthalath, Miss Anousone Phengsombut, Miss Viengnakhone Thongphachanh, Miss Toutana Hormkinkeo, Miss Bouakham Saiyphimchai, Mr Amkha Senethavysouk, Mr Somboun Nadonhai, Mr Xayasith Sengaroundeth, Miss Vilailad Phetlavanh, Miss Veosavanh Keovoravong, Miss Nongluck Xayyalath, Miss Maniphone Phimmaleen, Miss Anback Hongsivilay, Mr Lamngern Phodchanthonthavong who played vital roles in the project as inspectors in the Evaluation Pharmacy; Mr Somchai Chanthapany, Mr Sathaphone Bounmala, Mr Soulivong Souphanhthavong who conducted the Minilab analysis of the samples. We are very grateful to the Directors and staff of Mahosot Hospital for allowing us to install the Evaluation Pharmacy in the hospital grounds and to the late Dr Rattanaphone Phetsouvanh and to Assoc. Prof Mayfong Mayxay for their advice. We are grateful for the support of the Infectious Diseases Data Observatory and the Centre for Tropical Medicine & Global Health of Oxford University who invested in the administration of the project, and to the members of the Substandard and Falsified Medical Products Team of the World Health Organisation, Geneva. We are very grateful for the discussions with the manufacturers and developers of the devices and to Dr. Fred Behringer for conducting the UPLC analyses of the samples. We are very grateful for the technical assistance of Dr. Douglas Ball, Dr. Sonalini Khetrapal and Dr. Susann Roth.

## Author Contributions

**Conceptualization:** Céline Caillet, Serena Vickers, Stephen Zambrzycki, Vayouly Vidhamaly, Kem Boutsamay, Phonepasith Boupha, Paul N. Newton.

**Data curation:** Céline Caillet, Serena Vickers, Stephen Zambrzycki, Vayouly Vidhamaly, Kem Boutsamay, Phonepasith Boupha.

**Formal analysis:** Céline Caillet, Serena Vickers, Pimnara Peerawaranun, Mavuto Mukaka.

**Funding acquisition:** Paul N. Newton.

**Investigation:** Céline Caillet, Serena Vickers, Stephen Zambrzycki, Vayouly Vidhamaly, Kem Boutsamay, Phonepasith Boupha.

**Methodology:** Céline Caillet, Serena Vickers, Stephen Zambrzycki, Vayouly Vidhamaly, Kem Boutsamay, Phonepasith Boupha, Paul N. Newton.

**Project administration:** Céline Caillet, Serena Vickers, Paul N. Newton.

**Resources:** Céline Caillet, Serena Vickers, Stephen Zambrzycki, Vayouly Vidhamaly, Kem Boutsamay, Phonepasith Boupha.

**Supervision:** Céline Caillet, Serena Vickers, Paul N. Newton.

**Validation:** Céline Caillet, Serena Vickers, Pimnara Peerawaranun, Mavuto Mukaka.

**Visualization:** Céline Caillet, Serena Vickers, Vayouly Vidhamaly, Kem Boutsamay, Phonepasith Boupha, Paul N. Newton.

**Writing – original draft:** Céline Caillet.

**Writing – review & editing:** Serena Vickers, Stephen Zambrzycki, Facundo M. Fernández, Vayouly Vidhamaly, Kem Boutsamay, Phonepasith Boupha, Pimnara Peerawaranun, Mavuto Mukaka, Paul N. Newton.

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
