## [Decision Letter · Decision Letter 0]

8 Apr 2021

Dear Dr Caillet,

Thank you very much for submitting your manuscript "A comparative field evaluation of six medicine quality screening devices in Laos" for consideration at PLOS Neglected Tropical Diseases. As with all papers reviewed by the journal, your manuscript was reviewed by members of the editorial board and by several independent reviewers. In light of the reviews (below this email), we would like to invite the resubmission of a significantly-revised version that takes into account the reviewers' comments. 

The authors are advised to very carefully consider the reviewers' comments and suggestions if they decide to submit a revised manuscript for re-consideration.

We cannot make any decision about publication until we have seen the revised manuscript and your response to the reviewers' comments. Your revised manuscript is also likely to be sent to reviewers for further evaluation.

Sincerely,

Thuy Le

Associate Editor

Ricardo Fujiwara

Deputy Editor

The authors are advised to very carefully consider the reviewers' comments and suggestions if they decide to submit a revised manuscript for re-consideration.

Reviewer's Responses to Questions

**Key Review Criteria Required for Acceptance?**

**Methods**

-Are the objectives of the study clearly articulated with a clear testable hypothesis stated?

-Is the study design appropriate to address the stated objectives?

-Is the population clearly described and appropriate for the hypothesis being tested?

-Is the sample size sufficient to ensure adequate power to address the hypothesis being tested?

-Were correct statistical analysis used to support conclusions?

-Are there concerns about ethical or regulatory requirements being met?

Reviewer #1: The methods are well applied and the study design was appropriate. However, several general comments regarding the method may be pointed out:

1. The different devices are compared in the same way, which is, in my opinion, a misleading representation of their applicability. Indeed, the devices do not provide the same kind of information and should be used in different situations. For example, the PADs only provide information on the presence (only qualitative and not quantitative information) of a specific API and a few common excipients. This device cannot, and is not intended to (at least to the extent of my knowledge), identify specific brands or detecting substandards. Therefore, I think that the devices should be compared regarding their intended use. This should be clarified and maybe specified for each device.

2. In my opinion, the different results regarding the performances should be counterbalance by the advantages and drawback of the different techniques. Indeed, PADs have a poorer capacity at detecting falsified medicines but there is no calibration phase and their price is very low compared to most Raman and NIR devices. This may also be part of the final decision for the choice of a device. 

3. An important information is missing regarding the calibration and comparison algorithms used with the spectroscopic devices. This information may possibly be found elsewhere but it is not clear to the reader where to find it. However, this information is crucial when comparing the performances of the devices since a very wide panel of algorithms and calibration strategies may be envisaged. In addition, this calibration phase and the afferent difficulties should be emphasized since it constitutes a major limitation to the use of spectroscopic techniques. 

4. Have the statistical models (used to transform the spectral information into a PASS/FAIL result) been validated? If yes, this information together with the validation results should be provided.

Reviewer #2: The objectives were clearly articulated and the pilot study design was appropriate. The sample size should have been larger to increase statistical power. Also the authors do not describe sufficiently and clearly how the 'simulated' medicine samples were prepared and verified against their original products especially in term of their packaging and labeling. The study should have included more dosage forms, including tablet, capsule, injectable, suspension, etc. Overall, the study method is quite creative!

Reviewer #3: Are the objectives of the study clearly articulated with a clear testable hypothesis stated? 5/10

-Is the study design appropriate to address the stated objectives? 5/10

-Is the population clearly described and appropriate for the hypothesis being tested? N/A

-Is the sample size sufficient to ensure adequate power to address the hypothesis being tested? N/A

-Were correct statistical analysis used to support conclusions? 4/10

-Are there concerns about ethical or regulatory requirements being met? N/A

**Results**

-Does the analysis presented match the analysis plan?

-Are the results clearly and completely presented?

-Are the figures (Tables, Images) of sufficient quality for clarity?

Reviewer #1: _Table 1: The microphazir is not a FT based NIR device but it rather use a Hadamard transform. It is therefore more comparable to the NIR-S-G1 than to a FT instrument (for more information see doi: 10.1177/0003702818809719)

_The authors should add a table of abbreviations

Reviewer #2: Results were clearly presented with appropriate graphical presentations

Reviewer #3: - Does the analysis presented match the analysis plan? 5/10

-Are the results clearly and completely presented? 7/10

-Are the figures (Tables, Images) of sufficient quality for clarity? 10/10

**Conclusions**

-Are the conclusions supported by the data presented?

-Are the limitations of analysis clearly described?

-Do the authors discuss how these data can be helpful to advance our understanding of the topic under study?

-Is public health relevance addressed?

Reviewer #1: _A supplementary limitation that should be listed is the fact that the tested device may not correspond to the actual devices due to software/hardware evolution.

 _Line 144: a risk when analysing tablets through the blisters is the impact of the latter on the final decision. Indeed, from the reviewer experience, a same brand of medicine may be blistered with plastics that exhibit a different spectral signature depending on the batches or providers. In addition, degradation of the blister (e.g. due to exposition to UV light) may distort the spectral signature and lead to wrong conclusion.

 _Table 2: In my opinion, it is risky to use laboratory-made tablets to calibrate a NIR device and use this calibration set to analyse industrially manufactured tablets. The differences in excipient origin/particle size but also the tablet shape, force of compression etc. will have an impact on the spectrum. This may lead to over optimistic results because the detection of the “falsified” or “substandard” tablet may be linked to these differences rather on the API strength. This kind of information should be present in the validation of the methods.

 _The authors should discuss the performances of the techniques regarding mono vs multi component formulations. Indeed, it is very hard for spectroscopic techniques to detect the absence of artemether in a lumefantrine/artemether formulation whether it is easier for colorimetric techniques.

Reviewer #2: The conclusions drawn from this pilot study emphasize the need for further investigations and studies in real life settings and proper validation of the technologies prior to their deployment for post-market surveillance by the regulatory agencies. The limitations of this pilot study are clearly explained. The study is very relevant to the current public health issues on substandard and falsified medicines that need to be addressed.

Reviewer #3: Are the conclusions supported by the data presented? 2/10

-Are the limitations of analysis clearly described? 8/10

-Do the authors discuss how these data can be helpful to advance our understanding of the topic under study? 8/10

-Is public health relevance addressed? 6/10

**Editorial and Data Presentation Modifications?**

Reviewer #1: _Line 34: please specify “near infrared” for the Phazir and NIR-S-G1 to avoid confusion with the “mid infrared “ 4500a device at line 35.

Reviewer #2: No

Reviewer #3: Please see comments to the authors

**Summary and General Comments**

Reviewer #1: The paper submitted by Caillet et al. describes the comparison of six devices to screen the quality of medicines. The paper is well written and easy to read despite the huge amount of data and results. The present paper is one of the first papers to try to have an objective evaluation and comparison of screening devices in real (or almost realistic) conditions. This should be emphasized and was well appreciated by the reviewer.

The paper deserves publication in PLOS NTD since it provides very useful information regarding the acceptability of these techniques by field inspectors. This kind of information is crucial when developing new devices or methods to ensure their correct application and acceptance by end-users. Another interesting finding is the impact that the use of screening devices has on the usual visual inspection leading to potentially sloppy inspection. Indeed, the screening devices should be considered as a supplementary tool to help inspectors detecting falsifications but not as the ultimate solution.

Reviewer #2: (No Response)

Reviewer #3: The objective of this original work is to evaluate the usefulness and ease of use of six portable screening devices in the hands of Lao medical inspectors for an inspection in a simulated evaluation pharmacy.

This is a topical issue in the fight against the use of counterfeit drugs in non-industrialized countries that affect both human and animal health.

The text is well written, the summary tables are clear and even the limitations of the study are extremely well presented.

No worries on the form.

I wonder about the options chosen by the authors to clarify, to promote decision support for the use of such or such control devices.

First, I would like to come back to the two objectives: usefulness and ease of use

Could we define the term usefulness. because if I understood correctly the objective is still to discern "the true" from the "false" for drugs in terms of sensitivity specificity and/or false positive, etc. ..... 

To summarize, are we evaluating the "machine" or the "assistant" or both? If I understand correctly, the last option is retained.

Then the question arises of the assistants and more exactly of their training in terms of calibration (intra, extra examiner variability) as recommended by the WHO. It is impossible to evaluate the performance of the devices if one biases from the beginning by not placing oneself in correct human resources conditions. Unless you place yourself in a pragmatic and not explanatory logic, which is not the case of your study.

The statistical analysis can be improved. We have a "machine" effect, an "examiner" effect (which varies in intra and inter) and a "Machine "X "Examiner" dependent effect.

You thus approach in a very precise way the time spent by acts which is not in itself on a priority evaluation grid (absent in the material and methods).

In the same way, approaching the perception or satisfaction of users in a "QALY study" manner is not logical. Here again, it is necessary to prepare a robust evaluation questionnaire beforehand.

Concerning the discussion and the conclusion, if I am a public health decision-maker, I remain uncertain.

The logic, like any comparative trial, would be to prioritize the products or group of products and produce operational recommendations. Otherwise, we miss our target.

My advice, but it is only advice, would be to restructure your article. The priority would be: What are in terms of sensitivity, specification, etc... the best or worst devices. And this, by placing you in a pragmatic logic, ie, a standard level of training, "basic" by describing the nature of training in the form of figure or table of synthesis. And therefore take out the criteria "time" and "perception" of the analysis to arrive at objective "machine" recommendations more or less hierarchical. The time factor could then be a secondary outcome or... a second article

PLOS authors have the option to publish the peer review history of their article (what does this mean?). If published, this will include your full peer review and any attached files.

Reviewer #1: No

Reviewer #2: Yes: Souly Phanouvong

Reviewer #3: No
---

## [Decision Letter · Decision Letter 1]

23 Jul 2021

Dear Dr Caillet,

We are pleased to inform you that your manuscript 'A comparative field evaluation of six medicine quality screening devices in Laos' has been provisionally accepted for publication in PLOS Neglected Tropical Diseases.

Best regards,

Thuy Le

Associate Editor

Ricardo Fujiwara

Deputy Editor

Reviewer's Responses to Questions

**Key Review Criteria Required for Acceptance?**

**Methods**

-Are the objectives of the study clearly articulated with a clear testable hypothesis stated?

-Is the study design appropriate to address the stated objectives?

-Is the population clearly described and appropriate for the hypothesis being tested?

-Is the sample size sufficient to ensure adequate power to address the hypothesis being tested?

-Were correct statistical analysis used to support conclusions?

-Are there concerns about ethical or regulatory requirements being met?

Reviewer #1: My comments are fully answered

Reviewer #2: The authors have addressed my comments/suggestions provided in the earlier version satisfactorily. I have further comments on this revised version.

**Results**

-Does the analysis presented match the analysis plan?

-Are the results clearly and completely presented?

-Are the figures (Tables, Images) of sufficient quality for clarity?

Reviewer #1: My comments are fully answered

Reviewer #2: Results are well-presented and figures are of sufficient quality for clarity.

**Conclusions**

-Are the conclusions supported by the data presented?

-Are the limitations of analysis clearly described?

-Do the authors discuss how these data can be helpful to advance our understanding of the topic under study?

-Is public health relevance addressed?

Reviewer #1: My comments are fully answered

Reviewer #2: Conclusions are supported by the data generated from the study. The limitations of the study are also well-described. The study results would fairly help policy decision makers, regulatory officials and program managers as well as public health personnel understand better the capability and limitations of the 6 field detection technologies and thus inform their decision in selecting which one(s) to be used.

**Editorial and Data Presentation Modifications?**

Reviewer #1: My comments are fully answered

Reviewer #2: No comment.

**Summary and General Comments**

Reviewer #1: The paper may be published in its revised version

Reviewer #2: Overall, the study findings would contribute to the literature and practices in the fight against substandard and falsified antimalarial and other medicine classes potentially.

PLOS authors have the option to publish the peer review history of their article (what does this mean?). If published, this will include your full peer review and any attached files.

Reviewer #1: No

Reviewer #2: No

---

## [Editor Report · Acceptance letter]

23 Aug 2021

Dear Dr Caillet,

We are delighted to inform you that your manuscript, "A comparative field evaluation of six medicine quality screening devices in Laos," has been formally accepted for publication in PLOS Neglected Tropical Diseases.

Best regards,

Shaden Kamhawi

co-Editor-in-Chief

Paul Brindley

co-Editor-in-Chief
